# Green credit policy and corporate excess cash holdings

**Jinhui Ning**[1]*, **Guiping Wang**[2], **Fengshan Xiong**[1], **Shi Yin**[1]*

1 School of Economics and Management, Hebei Agricultural University, Baoding, China, 2 School of Economics and Management, Tianjin Renai College, Tianjin, China

* ningjinhui1120@163.com (JN); shyshi0314@163.com (SY)

**Data Availability Statement:** All relevant data are within the paper and its Supporting information files.

**Funding:** This research received funding from the Hebei Provincial Department of Education University Humanities and Social Sciences

## Abstract

Green credit is changing industrial structure and corporate behavior, but little attention has been paid to the relationship between green credit and corporate cash management behavior. Based on the typical fact that the allocation of traditional bank credit funds is biased towards heavily polluting industries and the exogenous impact event of green credit policy, this paper takes A-share listed companies in China's capital market from 2008 to 2015 as samples, and uses the DID model to investigate the impact of green credit policy on excess cash holdings of heavily polluting enterprises. The findings indicate that the green credit policy has reduced the excessive cash holdings of heavily polluting enterprises, suggesting that it can correct the issue and align their cash holdings with the requirements of normal production and operations. The mechanism test demonstrates that the green credit policy can alleviate agency conflicts and influence enterprise cash holdings. Moreover, a cross-sectional investigation reveals that the inhibitory effect of the green credit policy on cash holdings is more pronounced in large-scale and state-owned enterprises compared to small-scale and non-state-owned enterprises. Finally, an analysis of the economic consequences reveals that the green credit policy indirectly enhances corporate value by reducing excessive cash holdings. Based on this, banks and financial institutions continue to treat the credit granting of heavily polluting enterprises cautiously, optimize the structure of green financial products, fully consider the different types and nature of customers, and develop differentiated lending conditions and diversified evaluation mechanisms. This paper has enriched the research on the economic consequences of green credit and the influencing factors of corporate cash holdings, and provided policy enlightenment for regulators and listed companies to correctly understand and make full use of green credit policies to keep corporate cash stable through the crisis.

## 1. Introduction

In recent years, the increasingly severe environmental pollution and ecological imbalance have aroused widespread societal concern. China's environmental issues are closely related to the allocation of financial resources [1]. Under the traditional banking business model, heavily polluting enterprises easily obtain abundant credit cash flow due to the advantage of mortgage

Research Project "Research on Evaluation of Green Governance Effect of Environmental Pollution Liability Insurance for Listed Companies under the 'Double Carbon' Goal". The source of this funding is the Hebei Provincial Education Department. The funding grant number is BJS2022003. The funder (Jinhui Ning) had role in study design, data collection and analysis, decision to publish, or preparation of the manuscript.

**Competing interests:** The authors declare that the research was conducted in the absence of any commercial or financial relationships that could be construed as a potential conflict of interest.

assets [1, 2]. In addition, management blindly expands and strengthens the extensive development concept, which directly leads to low capital efficiency, distorted investment behavior, and serious environmental pollution (Chang et al., 2016; Luo et al., 2019) [3, 4]. In this context, our government attaches great importance to using green credit to reconcile the contradiction between economic development and environmental governance. Green credit emphasizes that financial institutions should incorporate the ecological environment into their investment and financing activities, allowing more funds to flow into green projects and limiting their flow into the pollution field, thereby changing the incentive mechanism for resource allocation (Ma et al., 2015) [5]. To this end, in 2012, the China Banking Regulatory Commission issued the "Green Credit Guidelines" (hereinafter referred to as the "Guidelines"), emphasizing that banks and financial institutions should fully consider environmental risks when lending, deny credit to customers whose environmental performance is not compliant, prevent environmental risks, and promote the green adjustment of economic structure. Not only does this guideline clearly define "green credit" in official documents for the first time, elevating green credit to the strategic height of banking institutions, but it also puts forward specific requirements for banking institutions from five aspects: organizational management, policy system and capacity building, process management, internal control management and information disclosure, and supervision and inspection. Major commercial banks have actively implemented green credit requirements. Industrial and Commercial Bank of China has formed an environmental assessment system for customer access and applied it to customer risk assessment. Bank of Communications has developed a "List of Environmental and Social Compliance Documents" based on loan investment, and China Construction Bank has released "Strengthening Credit Management for High Pollution and High Environmental Risk Enterprises". As an institutional innovation to strengthen the construction of ecological civilization, the green credit policy aims to limit the disorderly expansion of heavily polluting enterprises from the source of funds and to curb the spread of environmental pollution. Although heavily polluting enterprises are the main producers of environmental pollution, they are also the micro-entities of green development and high-quality development. So, can the green credit policy correct the cash decision making of heavily polluting enterprises, reduce inefficient capital behavior, make their cash holdings more reasonable, and better meet the needs of normal production and operation? In the context of green development, it is of great practical significance to study how to promote green credit policy in order to serve the high-quality capital operation of heavily polluting enterprises.

Cash management is an important component of corporate finance and business strategy research. The existing literature has found that management is more willing to hold excess cash for self-interest purposes [6–8]. As can be seen, cash, as the most liquid asset, is easily reduced to a tool for managers to seek personal gain. Therefore, if the optimal cash holding decision is made, preventing excess cash holdings is also a key decision for financial management in practical departments. Not only are cash holdings related to a company's operating behavior, but they also reflect a strategic response to the external financing environment (Opler et al., 1999; Denis and Sibilkov, 2010; Liu et al., 2017) [9–11]. The introduction of the Guidelines has provided a good economic scenario for examining the financial effects of green credit policy on heavily polluting enterprises. The existing literature on the factors affecting excess cash holdings in enterprises has found that transaction motivation (Mulligan, 1997) [12], preventive motivation (Opler et al., 1999) [9], financing constraints (Almeida et al., 2004) [13], corporate governance (Jensen, 1986; Xin and Xu, 2006) [6, 14], the personal characteristics of senior executives (Zhang and Huang, 2014) [15] and other internal and macroeconomic policy of the company (Lu and Han, 2013; Cai et al., 2015; Faulkender et al., 2019;Si and Li,2022) [16–19], the business environment (Wang et al., 2020) [20], and media reporting

tendencies (Zhi and Zhou, 2021) [21], and Annual report inquiry letter(Wang et al.,2022) [22], and other external aspects of the company can have a significant impact on the excess cash holdings of enterprises. However, the shaping role of the exogenous event of green credit policy in corporate cash decision making has been ignored. Based on this, this article attempts to take the "Guidelines" as a research opportunity to examine the complex relationship, functional path, cross-sectional differences, and economic consequences between green credit policy and the excess cash holdings of heavily polluting enterprises. The explanation of these issues is not only conducive to understanding the cash decision-making logic of heavily polluting enterprises but also provides reference significance for enterprises to more effectively use green credit policy to maintain corporate cash stability through crises.

Compared with the previous literature, the contributions of this article are mainly manifested in the following three aspects: Firstly, it enriches the literature on the evaluation of the effectiveness of green credit policy. The existing literature mainly evaluates the effectiveness of green credit policy from the perspectives of green innovation (Wang and Wang, 2021;Chen and Zheng,2023) [23, 24], investment efficiency (Ning et al., 2021) [25], Enterprise R&D investment [26] and financing scale (Cai et al., 2019) [27], Carbon reduction efficiency [28], corporate risk taking (Gao and Zhang, 2023) [29]. This article provides empirical evidence of the implementation effect of green credit policy more directly from the perspective of the excess cash holdings of enterprises, enriching the literature in the field of the evaluation of the effectiveness of green credit policy. Secondly, it expands the perspective of research on the influencing factors of corporate excess cash holdings. The driving factors of corporate cash holdings are an important topic of concern for scholars. Currently, the literature on the factors affecting corporate excess cash holdings is rich, but the literature on the impact of green credit policy on corporate excess cash holdings is scarce.

In the current context of the flourishing of green finance, green finance policy have an important impact on the external financing decisions of heavily polluting enterprises. Based on a quasi-natural experiment of the Guidelines, this article examines the impact, mechanism, cross-sectional differences, and economic consequences of green credit policy on the excess cash holdings of heavily polluting enterprises. While overcoming endogenous problems, it also expands the perspective of research on the factors affecting the excess cash holdings of enterprises. Thirdly, the research conclusions have an important practical value. This article examines the impact of green credit policy on the excess cash holdings of enterprises, providing a useful reference for the government on how to improve and revise green credit policy, better encourage heavily polluting enterprises to flexibly adjust their corporate cash decisions and implement sustainable development strategies to respond to changes and requirements in the external business environment, and fully utilize the positive externalities of green credit policy construction.

## 2. Theoretical analysis and hypothesis formulation

The principal–agent theory based on free cash flow believes that free cash flow is a product of conflicts of interest between management and external shareholders and that it is a manifestation of interest encroachment. When the external monitoring environment mechanism is relatively weak, management will not pay dividends to shareholders due to free cash flow considerations and is more willing to hold cash flows that exceed a reasonable level to achieve the goal of "building a business empire" or meeting on-the-job consumption (Jensen, 1986; Dittmar and Mahrt-Smith, 2007; Harford et al., 2008; Luo et al., 2018) [7, 14, 30, 31]. Although reasonable cash reserves are conducive to avoiding liquidity risks and promoting company performance improvement (Faulkender and Wang, 2006; Lyanderes and Palazzo,

2016) [32, 33], excessive cash holdings not only waste cash resources but also reduce cash value and company value (Harford, 1999; Ozkan and Ozkan, 2004; Guney et al., 2007) [34–36]. Overall, the more serious the corporate agency problem, the worse the internal governance mechanism, the higher the level of excess cash holdings, and the lower the value of the cash holdings (Ozkan and Ozkan, 2004; Guney et al., 2007) [35, 36]. Therefore, management agency conflicts are a key factor affecting the excess cash holdings of enterprises. If excessive cash holdings are the result of management's principal–agent problems, then solutions to mitigate agency conflicts can effectively reduce the level of cash holdings (Amess et al., 2015) [37].

The financing restrictions imposed by green credit policy on heavily polluting enterprises may weaken the agency motivation of their management, thereby reducing their excess cash holdings. Under the traditional banking business model, heavily polluting enterprises can easily obtain abundant credit cash flow due to the advantage of mortgage assets (Wei et al., 2023) [38]. This has led to a concentration of credit funds in basic key industries, such as capital-intensive, high energy consumption, and heavy pollution industries. In addition, management blindly expands and strengthens the extensive development concept, which directly leads to a low capital efficiency, distorted investment behavior, and serious environmental pollution (Chang et al., 2016; Luo et al., 2019) [3, 4]. The credit rationing theory believes that, in the face of excessive capital demand, banks should incorporate the corrosive effect of risk on profit quality into credit decision-making considerations; establish stricter lending conditions so that some capital demanders cannot obtain loans, even if they are willing to pay high interest rates; and, ultimately, withdraw from the bank loan market, eliminating excess demand to achieve credit balance. In essence, green credit is a credit rationing based on environmental constraints, that is, under the same capital price conditions, to tilt funds toward green projects, curb investment in polluting projects, and avoid a large number of low-level repetitive construction and environmental pollution through the allocation of credit resources in order to effectively guide enterprises to achieve intensive development. The Guidelines clearly state that "banking financial institutions should strengthen credit approval management and determine reasonable credit authorization and approval processes based on the nature and severity of environmental and social risks faced by customers. Customers with non compliant environmental and social performance should not be granted credit." In the context of emphasizing the construction of ecological civilization, the debt default risk of heavily polluting enterprises is increasing, and banks have adopted the measure of reducing loans. Measures such as loan deferral implement credit rationing for heavily polluting industries with high investment risks. The green credit policy will worsen the phenomenon of credit rationing for heavily polluting enterprises, significantly reducing the availability of credit, making it difficult to obtain bank loans, and straining cash flow. Many documents have found that green credit policy have reduced the credit scale of heavily polluting enterprises (Su and Lian, 2018; Cai, et al., 2019; Wang and Wang, 2021) [23, 27, 39]. The green credit policy has weakened the free cash flow of heavily polluting enterprises. The lower the funds available to management, the more prudent and reasonable the cash decisions that they make. Therefore, we believe that the green credit policy can reduce the agency excess cash holdings of heavily polluting enterprises, play a "corrective" role in their excess cash holdings, make their cash holdings more reasonable, and better meet the needs of normal production and operation. Based on the above analysis, this article proposes the following hypothesis:

H1: The green credit policy reduces the excess cash holdings of enterprises.

## 3. Research design

### 3.1. Sample selection and data sources

This article takes the issuance of the "Guidelines" as a research opportunity, selects Chinese A-share listed companies in Shanghai and Shenzhen as research samples, and empirically examines the impact of green credit policy on the excess cash holdings of enterprises. When selecting samples, this article processes the samples according to the following principles: (1) delete financial and insurance listed companies, (2) delete ST listed enterprises, and (3) exclude listed companies with missing financial data and corporate governance data; Our final sample includes 13 638 observations representing 2340 firms. We followed the relevant research (Su and Lian, 2018; Cai, et al., 2019; Wang and Wang, 2021) [23, 27, 39], and conducts a 1% quantile tail reduction process for all continuous variables in order to eliminate the impact of outliers on the empirical results. Financial data and corporate governance data are all from the CSMAR database, the WIND database, and the RESSET database.

### 3.2. Measurement of key variables

**3.2.1. Explained variable: Corporate excess cash holdings.**  The explanatory variable of this article is *Excess_Cash*. Referring to the practices of Opler et al. (1999) [9], Xin and Xu (2005) [6], Yang et al. (2010) [40], and Yu et al. (2019) [41], this article constructs the following corporate cash holding expectation model:

$$Cash_{i,t} = \alpha_0 + \alpha_1 Size_{i,t} + \alpha_2 CFO_{i,t} + \alpha_3 NETWC_{i,t} + \alpha_4 Growth_{i,t}$$
$$+ \alpha_5 Capex_{i,t} + \alpha_6 Lev_{i,t} + \alpha_7 Dividend_{i,t} + \varepsilon_{i,t} \tag{1}$$

*Cash* is the level of cash holdings, measured using the sum of cash and cash equivalents to measure the proportion of the difference between total assets and cash and cash equivalents. *Size* is the size of the enterprise, taken as the natural logarithm of the company's total assets. *CFO* is the operating cash flow, which is calculated by dividing the net cash flow generated from operating activities by non-cash assets. *NETWC* is net working capital, which is the difference between current assets and current liabilities divided by non-cash assets. *Growth* refers to growth, which is taken as the growth rate of sales revenue. *Capex* is the capital expenditure ratio, which is calculated by dividing the cash paid to obtain fixed assets, intangible assets, and other long-term assets by total assets. *Lev* is the financial leverage, taken as the asset liability ratio, the ratio of total liabilities to total assets. *Dividend* is a dummy variable for dividend payment. If the company announces dividend payments in the annual report of the current year, a value of 1 is assigned; otherwise, a value of 0 is assigned. The residual of model (1) represents the degree of deviation of the actual cash holdings of the enterprise from the expected cash holdings. The residual of model (1) is taken as an absolute value to measure the excess cash holdings. The smaller the indicator value, the lower the excess cash holdings of the enterprise and vice versa.

**3.2.2 Explanatory variable: Green credit policy.**  The explanatory variables in this article include group variables (*Treat*) and time variables (*Post*). *Treat* is a dummy variable set according to whether the enterprise is a heavily polluted enterprise, distinguishing the experimental group from the control group. This article refers to the practice of Pan et al. (2019) [42]. If the sample belongs to the mining industry, the textile services and fur industry, the metal and non-metal industry, the biomedical industry, the petrochemical and plastic industry, the paper and printing industry, the water and gas industry, and the food and beverage industry, it is defined as a heavily polluting enterprise, and the rest are defined as non-heavily polluting enterprises. In this paper, the value of the heavily polluted enterprises in the

experimental group is 1, and the value of the non-heavily polluting enterprises in the control group is 0. *Post* is a time variable of the green credit policy. Referring to the research by Wu et al. (2021) [43], this article uses the four years before and after the implementation of the green credit policy as the observation window period; that is, the four years before the implementation of the Guidelines (2008–2011) are examined, where the *Post* value is 0, and the four years after the implementation of the Guidelines (2012–2015) are examined, where the *Post* value is 1.

**3.2.3 Control variables.** Referring to the relevant literature on the influencing factors of corporate cash holdings, this article selects the following control variables: the company size (*Size*), financial leverage (*Lev*), return on total assets (*Roa*), property right nature (*Soe*), establishment period (*Age*), net cash flow generated from operating activities per share (*CF*), working capital ratio (*Ncap*), total asset turnover (*Turn*), board size (*Board*), total compensation of the top three executives (*Pay*), and shareholding ratio of the top ten shareholders (*Tthr*); this paper also controls for individual fixed effects and year fixed effects. See Table 1 for specific variable definitions and values.

## 3.3 Model design

In order to test the research hypothesis, this article refers to the practices in the existing literature (Bertrand and Mullainathan, 2003; Wang and Tan, 2019; Zeng et al., 2022) [44–46]. In order to alleviate the impact of unobserved corporate heterogeneity characteristics and unobservable characteristic factors that change over time on the research findings, the following panel bidirectional fixed effect DID model is established:

$$Excess\_Cash_{i,t} = \beta_0 + \beta_1 \times Post_{i,t} \times Treat_{i,t} + \beta_i \times Control + \mu_i + \mu_t + \xi_{i,t} \qquad (2)$$

where *Excess_Cash* is the explained variable of this article, *Post×Treat* is the explanatory

**Table 1. Variable definitions.**

| Variable Name | Variable symbol | Calculation method |
|---|---|---|
| Corporate excess cash holdings | *Excess_Cash* | Absolute value of residual in model (1). |
| Group variable | *Treat* | Whether it is a binary variable of a heavily polluted enterprise. If it is a heavily polluted enterprise, a value of 1 is assigned; otherwise, a value of 0 is assigned. |
| Time variable | *Post* | Binary variables before and after the implementation of the Guidelines are assigned a value of 1 in 2012 and beyond; otherwise, a value of 0 is assigned. |
| Company size | *Size* | Natural logarithm of total assets. |
| Financial leverage | *Lev* | Total liabilities divided by total assets. |
| Return on total assets | *Roa* | Net profit divided by total assets. |
| Nature of property rights | *Soe* | For state-owned enterprises, a value of 1 is assigned; otherwise, a value of 0 is assigned. |
| Years of establishment | *Age* | Natural logarithm of (current year—year of company establishment + 1). |
| Net cash flow from Operating activities per share | *CF* | Net cash flow from operating activities divided by total shares. |
| Working capital ratio | *Ncap* | Current assets minus current liabilities divided by total assets. |
| Total asset turnover rate | *Turn* | Operating revenue divided by total assets. |
| Board size | *Board* | Natural logarithm of the total number of directors. |
| Total compensation of top three executives | *Pay* | The natural logarithm of the total remuneration of the top three executives. |
| Shareholding ratio of top ten shareholders | *Tthr* | The total number of shares held by the top ten shareholders divided by the total number of shares. |

variable of this article, and *Control* is a control variable. $\beta_0$ is the intercept term, $\mu_i$ is the individual fixed effect, and $\mu_t$ is the year fixed effect. $\xi_{i,t}$ is the residual item, and $i$ and $t$ are the enterprise and time, respectively. This article focuses on coefficient $\beta_1$, which represents the change in the excess cash holdings of heavily polluting enterprises in the experimental group after the implementation of the green credit policy. If the estimated coefficient is significantly negative, it indicates that the green credit policy has reduced the excess cash holdings of heavily polluting enterprises.

## 4. Empirical results and analysis

### 4.1 Descriptive statistics

Table 2 reports the descriptive statistical results for the main variables. The average value of the excess cash holdings (*Excess_Cash*) of the enterprises is 0.0943, the standard deviation is 0.086, the minimum value is 0.00, and the maximum value is 0.44, indicating that there are significant differences in excess cash holdings among the listed companies in China. The average value of *Post* is 0.6568, and the samples after the issuance of the Guidelines account for approximately 65.68% of the total sample size. The average value of Treat is 0.2607, and the sample of heavily polluted enterprises accounts for approximately 26.07% of the total sample size. The descriptive statistical results of the other variables are basically in line with those in the previous literature, with no significant differences, and they are not shown here.

### 4.2 Basic regression results and analysis

Table 3 reports the basic regression results for the green credit policy and the excess cash holdings of enterprises. In column (1), the estimated coefficient of the green credit policy (*Post×Treat*) is -0.0148, and it is significantly negative at the statistical level of 1%. In column (2), after controlling for the other influencing factors, the estimated coefficient of the green credit policy (*Post×Treat*) is -0.0098, and it is significantly negative at the statistical level of 5%, which means that the green credit policy reduces the excess cash holdings of enterprises, meeting our theoretical expectations. This shows that green credit policy greatly reduces the credit availability of heavy polluting enterprises, making it difficult for them to obtain bank loans, resulting in tight cash flow, alleviating agency conflicts to a certain extent, and playing a role in

**Table 2. Descriptive statistics of main variables.**

| Variable Name | Number of samples | Average value | Standard deviation | Minimum value | Median | Maximum |
|---|---|---|---|---|---|---|
| *Excess_Cash* | 13638 | 0.0943 | 0.086 | 0.00 | 0.07 | 0.44 |
| *Post* | 13638 | 0.6568 | 0.475 | 0.00 | 1.00 | 1.00 |
| *Treat* | 13638 | 0.2607 | 0.439 | 0.00 | 0.00 | 1.00 |
| *Size* | 13638 | 22.1369 | 1.275 | 19.93 | 21.94 | 26.06 |
| *Lev* | 13638 | 0.4512 | 0.206 | 0.05 | 0.46 | 0.86 |
| *Roa* | 13638 | 0.0447 | 0.042 | -0.06 | 0.04 | 0.19 |
| *Tthr* | 13638 | 57.9015 | 15.457 | 22.32 | 58.95 | 90.28 |
| *Soe* | 13638 | 0.4801 | 0.500 | 0.00 | 0.00 | 1.00 |
| *Age* | 13638 | 2.7001 | 0.366 | 1.61 | 2.77 | 3.33 |
| *CF* | 13638 | 0.4164 | 0.831 | -2.30 | 0.32 | 3.68 |
| *Pay* | 13638 | 14.1382 | 0.702 | 12.37 | 14.13 | 16.04 |
| *Ncap* | 13638 | 0.2037 | 0.247 | -0.34 | 0.19 | 0.78 |
| *Board* | 13638 | 2.2781 | 0.178 | 1.79 | 2.30 | 2.77 |
| *Turn* | 13638 | 0.6558 | 0.459 | 0.08 | 0.54 | 2.62 |

**Table 3. Basic regression results.**

| Variable Name | Excess_Cash(1) | Excess_Cash(2) |
|:---:|:---:|:---:|
| Post × Treat | -0.0148*** | -0.0098** |
| | (-3.19) | (-2.11) |
| Size | | -0.0059* |
| | | (-1.72) |
| Lev | | 0.1588*** |
| | | (10.15) |
| Roa | | 0.0655** |
| | | (2.01) |
| Tthr | | -0.0006*** |
| | | (-3.84) |
| Soe | | -0.0021 |
| | | (-0.20) |
| Age | | 0.0032 |
| | | (0.19) |
| CF | | 0.0048*** |
| | | (4.10) |
| Pay | | 0.0036* |
| | | (1.87) |
| Ncap | | 0.1464*** |
| | | (13.10) |
| Board | | -0.0025 |
| | | (-0.20) |
| Turn | | -0.0106** |
| | | (-2.18) |
| Company FE | Yes | Yes |
| Year FE | Yes | Yes |
| _ Cons | 0.0751*** | 0.0938 |
| | (20.92) | (1.12) |
| N | 13638 | 13638 |
| R2 | 0.0073 | 0.0259 |
| F | 9.1708 | 15.0171 |
| P | 0.0000 | 0.0000 |

Note: The data in the table was calculated by the author using stata. The values in parentheses are t; * * *** And * represent significance at the 1%, 5%, and 10% levels, respectively.

"correcting" excess cash holdings of heavy polluting enterprises, making their cash holdings more reasonable and more in line with the needs of normal production and operation. It fits our theoretical expectations.

## 4.3 Robustness test

**4.3.1 Parallel trend hypothesis.** The important prerequisite for the establishment of the previous research conclusion is that the experimental group of heavily polluting enterprises and the control group of non-heavily polluting enterprises had parallel trends before the implementation of the Guidelines. This article refers to the parallel trend test method of Wu et al. (2021) [43], and it combines the dual difference model and event study method to test

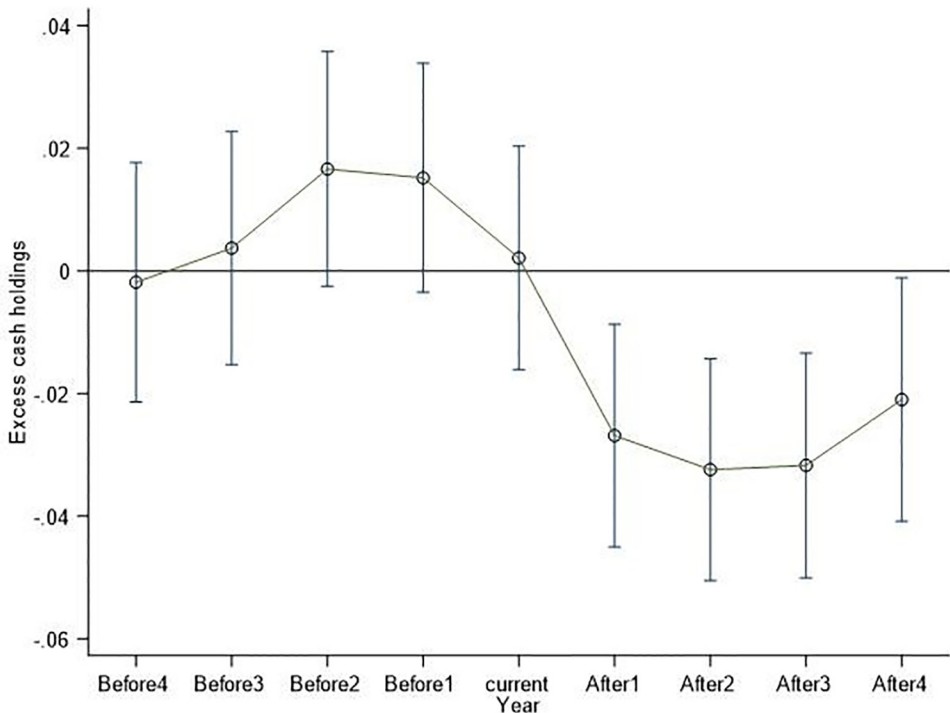

**Fig 1. Change trend of green credit policy before and after implementation.**

whether the sample in this article meets the sample trend hypothesis, as shown in Fig 1. Current year represents the year in which the policy is implemented. Before1, Before2, Before3, and Before4 represent the year before and the first two years, the first three years, and the first four years before the policy implementation, respectively. After1, After2, After3, and After4 represent the first year, the first two years, the third year, and the fourth year after the policy implementation, respectively. The results show that the sample policy in this article passed the parallel trend test required for the dual difference model estimation before implementation, and green credit played a role in suppressing the excessive cash holdings of heavily polluting enterprises after the implementation of the policy.

**4.3.2 Placebo test.** In this paper, the implementation time of the green credit policy is artificially moved forward one year; that is, it is assumed that the "Guidelines" event occurred in 2011, and then the model (1) is reused for regression. Table 4 reports the regression results of the placebo test. After adjusting the year in which the Guidance event occurs by one year ahead, the estimated coefficient of the cross-product term (*Post×Treat*) is 0.0056, which fails the significance test. This indicates that the decrease in the excess cash holdings of enterprises is indeed caused by the green credit policy event.

**4.3.3 Random construction of an experimental group and a control group.** In order to avoid the impact of the differences in the samples themselves on the research findings, this article randomly selects some samples as experimental groups, while the remaining samples are used as control groups. A new experimental group and a new control group (*Treat_random*) are constructed; that is, they are dummy variables of heavily polluted enterprises. Then, instead of the original Treatment variable, this new variable is used in model (1), and the regression estimation is repeated 500 times. The result is shown in Fig 2. The proportion of

**Table 4. Placebo test.**

| Variable Name | Excess_Cash |
|---|---|
| Post×Treat | 0.0056 |
| | (1.19) |
| Size | -0.0196*** |
| | (-5.96) |
| Lev | 0.3369*** |
| | (22.37) |
| Roa | 0.1375*** |
| | (4.38) |
| Tthr | 0.0000 |
| | (0.16) |
| Soe | -0.0002 |
| | (-0.02) |
| Age | -0.0589*** |
| | (-3.66) |
| CF | 0.0262*** |
| | (23.09) |
| Pay | 0.0019 |
| | (1.04) |
| Ncap | 0.5993*** |
| | (55.72) |
| Board | -0.0144 |
| | (-1.22) |
| Turn | -0.0133*** |
| | (-2.84) |
| Company FE | Yes |
| Year FE | Yes |
| _Cons | 0.4973*** |
| | (6.17) |
| N | 13638 |
| R2 | 0.2906 |
| F | 231.0688 |
| P | 0.0000 |

Note: The data in the table was calculated by the author using stata. The values in parentheses are t; * * *** And * represent significance at the 1%, 5%, and 10% levels, respectively.

regression coefficients of cross-product terms (*Post×Treat*) that are significantly positive and significantly negative is relatively small, supporting the findings of this study that the reduction in the excess cash holdings of heavily polluting enterprises is caused by green credit policy.

**4.3.4 Propensity score matching test.** Considering that there are systematic differences between the experimental group and the control group before the event shock, which may lead to sample selection bias, this article further uses propensity score matching to match and screen the regression samples so that there is no significant difference between the two groups of samples regarding the basic characteristics of the companies. Referring to the practice of Cheng et al. (2021) [47], this article uses Treatment as the explanatory variable, and all control variables in model (1) are used as the explanatory variables. A logit regression model is used to calculate the propensity score, and then a 1:1 ratio of nearest neighbor matching is used to pair

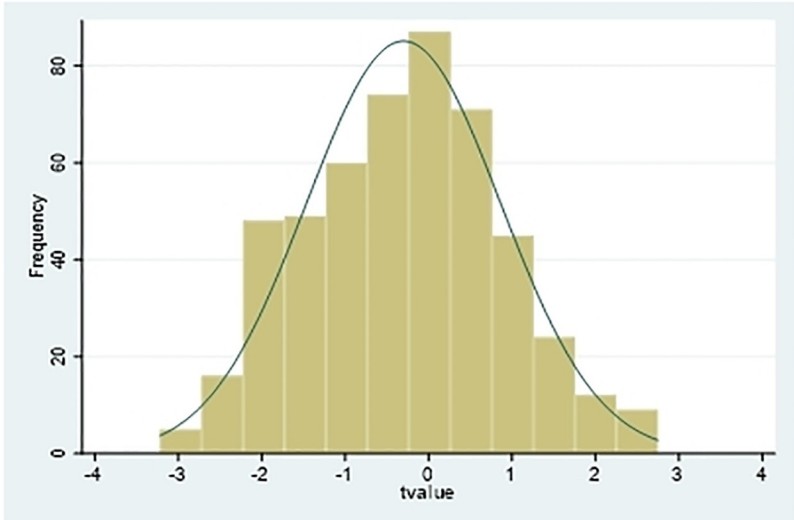

**Fig 2. Placebo test.**

non-return samples. Finally, the matched samples are used to regress model (1) again. Table 5 reports the regression results of the propensity score matching sample. The estimated coefficients of the cross-product terms (*Post×Treat*) are -0.0186 and -0.0121, respectively, which are significantly negative at the 1% statistical level. The above shows that after controlling sample selection bias, the implementation of green credit policy does reduce the cash holding level of enterprises, The test results are consistent with the previous results, demonstrating the robustness of the research findings.

**4.3.5 Selection criteria for replacing heavily polluting enterprises.** In order to verify the robustness of the industry attribute indicators of Treatment's heavily polluting enterprises, this article further draws on the practices of Liu and Liu (2015) [48], and it uses the following industries to define the scope of the experimental group of heavy pollution industries: the oil and gas mining industry; the ferrous metal mining and dressing industry; the nonferrous metal mining and dressing industry; the petroleum processing, coking, and nuclear fuel processing industry; the chemical raw materials and chemical products manufacturing industry; the chemical fiber manufacturing industry; the rubber and plastic products industry; the nonmetallic mineral products industry; the ferrous metal smelting and rolling processing industry; the nonferrous metal smelting and rolling processing industry; and the electric power, thermal production, and supply industries. Samples from the other industries are used as control groups. Table 6 reports the empirical results obtained after replacing the selection criteria for heavily polluting enterprises. The estimated coefficients for column (1) and column (2) are -0.0212 and -0.0102, respectively, which are significantly negative at the 1% statistical level. Thus, it strengthens the influence of green credit policy on excess cash holdings of enterprises, and further proves that the empirical results of this paper are still robust.

**4.3.6 Change the enterprise's excess cash holding variables.** In order to ensure the robustness of the enterprise's excess cash holdings index, reference is made to the practice of Luo et al. (2018) [31] to measure the excess cash holdings of enterprises by subtracting the cash holdings of the same industry. Table 7 reports the test results obtained after replacing the variable of the excess cash holdings of enterprises. The estimated coefficients of the cross-product terms (*Post×Treat*) of column (1) and column (2) are -0.0879 and -0.0278, which are

**Table 5. Results of propensity score matching test.**

| Variable Name | Excess_Cash(1) | Excess_Cash(2) |
|---|---|---|
| *Post × Treat* | -0.0186*** | -0.0121*** |
| | (-6.59) | (-2.78) |
| *Size* | | -0.0108*** |
| | | (-2.98) |
| *Lev* | | 0.0957*** |
| | | (6.14) |
| *Roa* | | 0.0200 |
| | | (0.68) |
| *Tthr* | | -0.0002 |
| | | (-1.56) |
| *Soe* | | -0.0044 |
| | | (-0.38) |
| *Age* | | -0.0048 |
| | | (-0.29) |
| *CF* | | 0.0029** |
| | | (2.08) |
| *Pay* | | 0.0024 |
| | | (1.12) |
| *Ncap* | | 0.0955*** |
| | | (8.25) |
| *Board* | | 0.0067 |
| | | (0.56) |
| *Turn* | | -0.0040 |
| | | (-0.86) |
| Company FE | Yes | Yes |
| Year FE | Yes | Yes |
| _ Cons | 0.0666*** | 0.2307*** |
| | (18.80) | (2.72) |
| N | 5187 | 5187 |
| R2 | 0.0141 | 0.0333 |
| F | 8.2472 | 6.4792 |
| P | 0.0000 | 0.0000 |

Note: The data in the table was calculated by the author using stata. The values in parentheses are t; * * *** And * represent significance at the 1%, 5%, and 10% levels, respectively.

significantly negative at the statistical levels of 1% and 5%, respectively, indicating that the conclusions in this paper have some robustness.

**4.3.7 Change the sample inspection period.** We changed the testing period from 4 years before and after the implementation of the Guidelines to 3 years, and we conducted a regression on model (1). Table 8 reports the regression results for the changed sample testing period. The estimated coefficients of the intersection and multiplication terms are -0.0083 and -0.0067, respectively, which are significantly negative at the statistical level of 5%. The research findings are unaffected.

**4.3.8 Replacement regression model.** This article further uses the least squares OLS model for validation. Table 9 reports the empirical results of the alternative regression model. The estimated coefficients of the cross-products of column (1) and column (2) are -0.0111 and

**Table 6. Selection criteria for replacement of heavily polluting enterprises.**

| Variable Name | Excess_Cash(1) | Excess_Cash(2) |
|:---:|:---:|:---:|
| Post × Treat | -0.0212*** | -0.0102*** |
|  | (-6.87) | (-3.22) |
| Size |  | -0.0072*** |
|  |  | (-6.43) |
| Lev |  | 0.0748*** |
|  |  | (9.69) |
| Roa |  | 0.0111 |
|  |  | (0.42) |
| Tthr |  | -0.0001 |
|  |  | (-0.91) |
| Soe |  | 0.0037 |
|  |  | (1.57) |
| Age |  | 0.0161*** |
|  |  | (5.37) |
| CF |  | 0.0067*** |
|  |  | (6.19) |
| Pay |  | 0.0032** |
|  |  | (2.43) |
| Ncap |  | 0.0909*** |
|  |  | (15.64) |
| Board |  | -0.0163*** |
|  |  | (-2.60) |
| Turn |  | 0.0092*** |
|  |  | (4.86) |
| Company FE | Yes | Yes |
| Year FE | Yes | Yes |
| _ Cons | 0.0993*** | 0.1458*** |
|  | (86.62) | (5.67) |
| N | 13638 | 13638 |
| R2 | 0.0034 | 0.0314 |
| F | 47.1366 | 36.8514 |
| P | 0.0000 | 0.0000 |

Note: The data in the table was calculated by the author using stata. The values in parentheses are t; * * *** And * represent significance at the 1%, 5%, and 10% levels, respectively.

-0.0064, respectively, both significantly negative at the statistical level of 5%, indicating that the green credit policy reduces the excess cash holdings of enterprises, further demonstrating the robustness of this conclusion.

## 4.4 Discussion

**4.4.1 Test of action mechanism based.** On the previous theoretical analysis, this article anticipates that green credit policy can play a "corrective" role in the excess cash holdings of heavily polluting enterprises by alleviating management agency conflicts. The chain of action is "green credit policy—agency costs—excess cash holdings of enterprises"; that is, reducing agency costs is the intermediary factor that enables green credit policy to affect the excess cash

**Table 7. Replacing the variable of excess cash holdings of enterprises.**

| Variable Name | Excess_Cash(1) | Excess_Cash(2) |
|---|---|---|
| *Post × Treat* | -0.0879*** | -0.0278** |
| | (-6.50) | (-2.11) |
| *Size* | | -0.0150*** |
| | | (-3.02) |
| *Lev* | | -0.0421 |
| | | (-1.23) |
| *Roa* | | -0.1285 |
| | | (-1.10) |
| *Tthr* | | 0.0009*** |
| | | (2.63) |
| *Soe* | | 0.0080 |
| | | (0.76) |
| *Age* | | -0.0292** |
| | | (-2.20) |
| *CF* | | 0.0306*** |
| | | (6.40) |
| *Pay* | | -0.0060 |
| | | (-1.00) |
| *Ncap* | | 0.4165*** |
| | | (16.11) |
| *Board* | | 0.0282 |
| | | (1.01) |
| *Turn* | | -0.0095 |
| | | (-1.14) |
| Company FE | Yes | Yes |
| Year FE | Yes | Yes |
| _ Cons | 0.1503*** | 0.5209*** |
| | (8.50) | (4.57) |
| N | 13638 | 13638 |
| R2 | 0.0121 | 0.0476 |
| F | 18.4887 | 56.7213 |
| P | 0.0000 | 0.0000 |

Note: The data in the table was calculated by the author using stata. The values in parentheses are t; * * *** And *
represent significance at the 1%, 5%, and 10% levels, respectively.

holdings of enterprises. To test this potential mechanism, this article draws on the mediation
effect model proposed by Imai et al. (2010) [49] and Wen et al. (2004) [50]. The first step is
consistent with model (2), the second step is shown in model (3), and the third step is shown
in model (4). By combining model (2), the following mediation effect model is constructed:

$$Agc_{i,t} = \alpha_0 + \alpha_1 \times Post_{i,t} \times Treat_{i,t} + \alpha_i \times Control + \mu_i + \mu_t + \xi_{i,t} \tag{3}$$

$$Excess\_Cash_{i,t} = \gamma_0 + \gamma_1 \times Post_{i,t} \times Treat_{i,t} + \gamma_2 \times Agc_{i,t} + \gamma_i \times Control + \mu_i + \mu_t + \xi_{i,t} \tag{4}$$

where *Agc* is a mediation variable. Drawing on the common practice of the literature related to
corporate governance, this article uses the management expense ratio to measure corporate

**Table 8. Change the sample inspection period.**

| Variable Name | Excess_Cash(1) | Excess_Cash(2) |
|:---:|:---:|:---:|
| $Post \times Treat$ | -0.0083** | -0.0067** |
| | (-2.57) | (-2.11) |
| Size | | -0.0111*** |
| | | (-4.01) |
| Lev | | 0.0889*** |
| | | (7.58) |
| Roa | | 0.0440* |
| | | (1.87) |
| Tthr | | -0.0002** |
| | | (-2.02) |
| Soe | | 0.0000 |
| | | (0.00) |
| Age | | 0.0154 |
| | | (1.24) |
| CF | | 0.0063*** |
| | | (6.89) |
| Pay | | 0.0019 |
| | | (1.37) |
| Ncap | | 0.1044*** |
| | | (12.81) |
| Board | | 0.0081 |
| | | (0.88) |
| Turn | | -0.0049 |
| | | (-1.33) |
| Company FE | Yes | Yes |
| Year FE | Yes | Yes |
| _ Cons | 0.0904*** | 0.1979*** |
| | (41.91) | (2.97) |
| N | 11501 | 11501 |
| R2 | 0.0048 | 0.0291 |
| F | 6.3619 | 15.3465 |
| P | 0.0000 | 0.0000 |

Note: The data in the table was calculated by the author using stata. The values in parentheses are t; * * *** And * represent significance at the 1%, 5%, and 10% levels, respectively.

agency issues. The higher the overhead rate, the more serious the agency problem. Table 10 reports the results of the mediation effect test. The estimated coefficient of the cross-product term (*Post×Treat*) of column (1) is -0.0062, which is significantly negative at the statistical level of 1%. The estimated coefficient of the cross-product (*Post×Treat*) in column (2) is -0.0145, which is significantly negative at the statistical level of 1%. The estimated coefficient of the agency problem (*Agc*) is 0.0459, which is significantly positive at the statistical level of 1%, indicating that mitigating agency conflicts plays a partial intermediary effect between green credit policy and the excess cash holdings of enterprises; On the one hand, green credit policy alleviates the free cash flow of heavily polluting enterprises and reduces the funds available to management, thus alleviating agency conflicts. On the other hand, management agency conflict is a key factor affecting excess cash holdings. Reducing agency conflicts can restrain excess

**Table 9. Replacement regression model.**

| Variable Name | Excess_Cash(1) | Excess_Cash(2) |
|:---:|:---:|:---:|
| Post × Treat | -0.0111** | -0.0064** |
| | (-2.25) | (-1.96) |
| Treat | -0.0081 | -0.0001 |
| | (-1.53) | (-0.03) |
| Size | | -0.0050*** |
| | | (-6.49) |
| Lev | | 0.0790*** |
| | | (13.80) |
| Roa | | 0.0020 |
| | | (1.26) |
| Tthr | | -0.0000 |
| | | (-0.45) |
| Soe | | -0.0004 |
| | | (-0.24) |
| Age | | 0.0144*** |
| | | (6.78) |
| CF | | 0.0061*** |
| | | (8.70) |
| Pay | | 0.0012 |
| | | (1.35) |
| Ncap | | 0.0991*** |
| | | (22.35) |
| Board | | -0.0049 |
| | | (-1.18) |
| Turn | | -0.0006 |
| | | (-0.46) |
| Company FE | Yes | Yes |
| Year FE | Yes | Yes |
| _ Cons | 0.1003*** | 0.1043*** |
| | (64.73) | (5.62) |
| N | 13638 | 13638 |
| R2 | 0.0628 | 0.1498 |
| F | 8.0916 | 51.6557 |
| P | 0.0003 | 0.0000 |

Note: The data in the table was calculated by the author using stata. The values in parentheses are t; * * *** And * represent significance at the 1%, 5%, and 10% levels, respectively.

cash holdings and make managers' cash decisions more prudent and reasonable. This is completely consistent with the theoretical analysis in this paper.

**4.4.2 Cross-section inspection.** Before the implementation of the green credit policy, large-scale heavily polluting enterprises were more likely to obtain credit funds due to having a lower information asymmetry, a strong repayment ability, and the advantage of mortgage assets. Therefore, the research by Su and Lian (2018) found that the green credit policy has a stronger credit penalty effect on large-scale heavily polluting enterprises [39]. In this paper, the total sample is divided into two groups according to the size of the enterprise, namely, small-scale enterprises and large-scale enterprises. The regression results are shown in Table 11.

**Table 10. Mediation effect test.**

| Variable Name | Agc(1) | Excess_Cash(2) |
|---|---|---|
| Post × Treat | -0.0062*** | -0.0145*** |
| | (-2.80) | (-6.72) |
| Agc | | 0.0459*** |
| | | (6.73) |
| Size | | -0.0064*** |
| | | (-8.07) |
| Lev | | 0.0629*** |
| | | (11.67) |
| Roa | | 0.0171 |
| | | (0.95) |
| Tthr | | -0.0000 |
| | | (-0.55) |
| Soe | | 0.0053*** |
| | | (3.20) |
| Age | | 0.0110*** |
| | | (5.13) |
| CF | | 0.0067*** |
| | | (9.20) |
| Pay | | 0.0017* |
| | | (1.87) |
| Ncap | | 0.0767*** |
| | | (19.31) |
| Board | | -0.0061 |
| | | (-1.43) |
| Turn | | 0.0124*** |
| | | (9.47) |
| Company FE | Yes | Yes |
| Year FE | Yes | Yes |
| _ Cons | 0.1419*** | 0.1160*** |
| | (82.81) | (6.30) |
| N | 13638 | 13638 |
| R2 | 0.0317 | 0.0679 |
| F | 41.0869 | 47.2018 |
| P | 0.0000 | 0.0000 |

Note: The data in the table was calculated by the author using stata. The values in parentheses are t; * * *** And * represent significance at the 1%, 5%, and 10% levels, respectively.

Column (1) reports the regression results for a sample group of small businesses. The estimated coefficient of Post×Treat is -0.0036, which is not significant. Column (2) reports the regression results for a sample group of large-scale enterprises. The estimated coefficient of Post×Treat is -0.0086, which is significantly negative at the statistical level of 5%. The inter-group coefficient difference test showed that the coefficient difference between the two groups of samples is significant at the level of 1%. This indicates that, compared to small-scale enterprises, the inhibitory effect of green credit policy on the excess cash holdings of enterprises is more obvious in large-scale enterprises. The possible reasons are that large-scale heavy polluting enterprises are often involved in multiple industries and fields, and their project

**Table 11. Cross-section inspection results.**

| Variable Name | Small-scale enterprises | Large-scale enterprises | Non-state-owned enterprises | State-owned enterprises |
|---|---|---|---|---|
| Post × Treat | -0.0036 | -0.0086** | -0.0062 | -0.0106*** |
| | (-0.87) | (-2.19) | (-1.40) | (-2.83) |
| Size | | | -0.0028** | -0.0077*** |
| | | | (-1.98) | (-7.63) |
| Lev | 0.0892*** | 0.0711*** | 0.1138*** | 0.0597*** |
| | (9.99) | (9.06) | (11.96) | (7.54) |
| Roa | 0.0030 | 0.0454* | 0.0063 | 0.0450 |
| | (0.11) | (1.72) | (0.25) | (1.63) |
| Tthr | -0.0000 | 0.0000 | -0.0002** | 0.0003*** |
| | (-0.33) | (0.45) | (-2.09) | (3.47) |
| Soe | -0.0004 | -0.0035 | | |
| | (-0.14) | (-1.59) | | |
| Age | 0.0129*** | 0.0156*** | 0.0116*** | 0.0195*** |
| | (3.92) | (5.07) | (3.97) | (5.33) |
| CF | 0.0115*** | 0.0039*** | 0.0055*** | 0.0062*** |
| | (6.38) | (5.07) | (4.04) | (7.21) |
| Pay | -0.0034** | 0.0015 | -0.0003 | 0.0016 |
| | (-2.47) | (1.24) | (-0.23) | (1.24) |
| Ncap | 0.1277*** | 0.0843*** | 0.1316*** | 0.0770*** |
| | (18.48) | (13.05) | (18.86) | (11.88) |
| Board | -0.0124* | -0.0023 | -0.0215*** | 0.0095 |
| | (-1.81) | (-0.42) | (-3.27) | (1.64) |
| Turn | -0.0040* | 0.0000 | -0.0007 | -0.0001 |
| | (-1.80) | (0.00) | (-0.33) | (-0.05) |
| Company FE | Yes | Yes | Yes | Yes |
| Year FE | Yes | Yes | Yes | Yes |
| _ Cons | 0.0726*** | -0.0192 | 0.1095*** | 0.1067*** |
| | (2.82) | (-0.84) | (3.51) | (3.99) |
| N | 6899 | 6737 | 7091 | 6547 |
| R2 | 0.1389 | 0.1762 | 0.1465 | 0.1674 |
| F | 36.1576 | 22.7971 | 37.4419 | 26.7413 |
| P | 0.0000 | 0.0000 | 0.0000 | 0.0000 |
| Inter group coefficient Difference test | chi2 = 11.53 Prob>chi2 = 0.001 | | chi2 = 13.16 Prob>chi2 = 0.000 | |

Note: The data in the table was calculated by the author using stata. The values in parentheses are t; * * *** And * represent significance at the 1%, 5%, and 10% lev-els, respectively.

investment scale is generally large, requiring a large amount of financial support. However, green credit policies assess the environmental impact of enterprises when providing loans, and are more inclined to support enterprises and projects with good environmental protection. Therefore, the green credit policy has a stronger credit punishment effect on large-scale heavy polluting enterprises, and the credit funds are more difficult to obtain after the implementation of the policy, which further weakens the free cash flow of state-owned heavy polluting enterprises, and the management has less funds available, which makes the cash decisions made more prudent and reasonable, and plays an effect of correcting excess cash holdings.

Therefore, the inhibition effect of green credit policy on excess cash holding of enterprises is more obvious in large-scale enterprises.

"Patriarchy", "implicit guarantees", soft budget constraints, and special attributes make state-owned enterprises less constrained by financing, while non-state enterprises often face discrimination and cold-shoulder treatment in the credit market. In particular, before the implementation of the green credit policy, state-owned heavily polluting enterprises were more likely to obtain credit loans from financial institutions. Therefore, Su and Lian (2018) found that green credit has a stronger credit penalty effect on state-owned heavily polluting enterprises [39]. In this paper, the total sample is divided into two groups according to company attributes: state-owned enterprises and non-state enterprises. The regression results are shown in Table 11. Column (3) reports the regression results for a sample group of non-state-owned enterprises. The estimated coefficient of $Post{\times}Treat$ is -0.0062, which is not significant. Column (4) reports the regression results for a sample group of state-owned enterprises. The estimated coefficient of $Post{\times}Treat$ is -0.0106, which is significantly negative at the 1% significance level. The intergroup coefficient difference test showed that the coefficient difference between the two groups of samples is significant at the level of 1%. This indicates that, compared to non-state-owned enterprises, the inhibitory effect of green credit policy on the excess cash holdings of enterprises is more obvious in state-owned enterprises. The possible reasons are as follows: on the one hand, state-owned heavy polluters, as representatives of state property, are usually more affected by government policy orientation. The promulgation of green credit policies has clarified the orientation of the state in environmental protection and put more emphasis on green development and sustainable development, and state-owned enterprises are more likely to be guided and constrained by green credit policies. On the other hand, state-owned heavy polluters are faced with more public concern and social responsibility in the operation process. Once environmental problems occur, greater losses may be caused to the corporate image and reputation. Therefore, the green credit policy has a stronger credit punishment effect on state-owned heavy polluters, which further weakens the free cash flow of state-owned heavy polluters. With less funds available to management, cash decisions made are more prudent and reasonable, and the inhibition effect of green credit policies on excess cash holdings is more obvious in state-owned enterprises.

## 4.5 Economic consequence test

Through the previous theoretical analysis and empirical testing, it is shown that the green credit policy significantly reduces the level of the excess cash holdings of enterprises, and it corrects the degree of deviation of the cash holdings from expectations. However, whether this impact can enhance enterprise value is an important issue that needs further analysis in this article. To this end, this paper constructs a multiplier between green credit policy and the excess cash holdings of enterprises ($After_{i,t}{\times}Treat_{i,t}{\times}Excess\_Cash_{i,t}$), and it uses *Tobin Q* to measure enterprise value. Table 12 reports the test results for economic consequences. The coefficient of the influence of the continuous multiplication term ($After_{i,t}{\times}Treat_{i,t}{\times}Excess\_Cash_{i,t}$) on enterprise value is 2.1975, which is significantly positive at the statistical level of 1%, indicating that the implementation of green credit policy can promote the improvement of corporate value by suppressing the excess cash holdings of heavily polluting enterprises. The possible reasons are that, on the one hand, the implementation of green credit policy encourages enterprises to transform to green economy, and by reducing excess cash holdings, enterprises can spend more funds on innovation and research and development of environmental protection technology, improve the environmental performance of products and services, and enhance the market competitiveness of enterprises. Enterprises can meet consumer demand

**Table 12. Economic consequence test.**

| Variable Name | Tobin Q |
|---|---|
| Post × Treat | -0.3833*** |
| | (-6.34) |
| Excess_Cash | -0.0143 |
| | (-0.14) |
| Post × Treat × Excess_Cash | 2.1975*** |
| | (4.80) |
| Size | -0.9358*** |
| | (-25.40) |
| Lev | -0.5704*** |
| | (-3.36) |
| Tthr | 0.0113*** |
| | (6.71) |
| Soe | -0.1636 |
| | (-1.51) |
| Age | 0.2612 |
| | (1.46) |
| CF | -0.0397*** |
| | (-3.15) |
| Pay | -0.0077 |
| | (-0.37) |
| Ncap | -0.8636*** |
| | (-7.11) |
| Board | 0.1410 |
| | (1.06) |
| Turn | 0.1782*** |
| | (3.39) |
| Roa | 9.1038*** |
| | (26.03) |
| Company FE | Yes |
| Year FE | Yes |
| _ Cons | 20.0160*** |
| | (22.18) |
| N | 13328 |
| R2 | 0.3187 |
| F | 233.3491 |
| P | 0.0000 |

Note: The data in the table was calculated by the author using stata. The values in parentheses are t; * * *** And * represent significance at the 1%, 5%, and 10% levels, respectively.

for environmentally friendly products and services, improve brand recognition and market share, and then increase the revenue and profit of enterprises, and enhance the value of enterprises. On the other hand, the implementation of green credit policy limits the excessive accumulation of cash by heavy polluting enterprises by restraining their excess cash holdings, and enterprises may use more funds for environmental protection investment and reduce environmental pollution risks. This helps to reduce the legal, administrative and reputational risks

faced by enterprises, improve the operating environment and business prospects of enterprises, and increase the value of enterprises.

## 5. Research conclusions and implications

This paper empirically examines the complex relationship, action path, cross-sectional differences, and economic consequences between green credit policy and the excess cash holdings of heavily polluting enterprises by employing the difference-in-differences(DID) model, using the exogenous impact of green credit policy. Through an empirical analysis, the following conclusions are drawn: (1) Green credit policy significantly reduces the level of the excess cash holdings of enterprises. (2) Green credit policy can alleviate agency conflicts and act on the excess cash holdings of enterprises. (3) Compared with small-scale and non-state-owned enterprises, the impact of green credit policy on the excess cash holdings of enterprises is mainly more obvious in large-scale and state-owned enterprises. (4) Green credit policy can play an indirect role in improving corporate value by reducing excess cash holdings. This article provides new insights for the in-depth understanding of the micro-capital transmission effects of green credit policy, enriches the performance evaluation of green credit policy and the research on the influencing factors of corporate cash holdings, and provides policy recommendations and inspiration for regulators and listed companies to correctly understand and fully utilize green credit policy to maintain corporate cash stability through crises.

Based on the above research conclusions, this article proposes the following countermeasures and suggestions: First, after the promulgation of the Guidelines, commercial banks strictly controlled the credit threshold for heavily polluting enterprises and reduced or even stopped providing bank loans to these enterprises, which played a "corrective" role in their management's agency excess cash holdings, making the cash holdings level more reasonable and more in line with the needs of normal production and operation. This means that the green credit policy has achieved good phased implementation results. Banking and financial institutions should continue to be cautious in granting credit to heavily polluting enterprises, continuously develop and optimize the structure of green financial products, and meet different customer needs. Secondly, the green credit policy only reduced the excess cash holdings of large-scale and state-owned heavily polluting enterprises, and it did not have a significant impact on small-scale and non-state-owned enterprises. This indicates that there is a significant "asymmetry" in the implementation of green credit policy. To this end, commercial banks should fully consider the different types and nature of customers; develop differentiated lending conditions and diversified evaluation mechanisms; and try to avoid the task-based implementation of overall quantitative objectives, which weaken the effectiveness of the policy implementation. Finally, although the implementation of the Guidelines has reduced the excess cash holdings of heavily polluting enterprises, the real purpose of the green credit policy is to force heavily polluting enterprises to phase out outdated production capacity and achieve intensive development through green technology innovation. The "one size fits all" policy will ultimately lead to the withdrawal of heavily polluting enterprises capable of or committed to green transformation and upgrading from the credit market, thereby departing from the original intention of the green credit policy. Therefore, while implementing credit restrictions on heavily polluting enterprises, banking institutions should also formulate corresponding incentive measures to increase financial support for environmentally friendly enterprises, and they should guide heavily polluting industries to contribute to developing green, circular, and low-carbon economies through leverage.

Although this paper proposes that green credit policies can inhibit excess cash holdings of enterprises, it only reveals that agency cost is the intermediary mechanism of its influence.

Subsequent studies can further explore other possible intermediary mechanisms. For example, based on the theoretical basis of this paper, the mediating role of variables such as information asymmetry and business risk can be explored in the future. In addition, this paper mainly explores the regulatory effects of firm size and property rights on green credit policies and excess cash holdings, and other possible boundary conditions can be explored in the future. For example, in view of external governance factors, we can explore the impact of factors such as the degree of public environmental concern and media supervision. In view of the internal governance factors, we can explore the moderating effect of managers' risk attitude and internal control.

## Supporting information

**S1 Dataset.**
(XLS)

## Author Contributions

**Conceptualization:** Guiping Wang.

**Data curation:** Jinhui Ning.

**Formal analysis:** Jinhui Ning.

**Investigation:** Fengshan Xiong.

**Project administration:** Shi Yin.

**Resources:** Guiping Wang.

**Software:** Guiping Wang, Fengshan Xiong.

**Validation:** Fengshan Xiong.

**Writing – original draft:** Jinhui Ning, Guiping Wang, Shi Yin.

**Writing – review & editing:** Jinhui Ning, Guiping Wang, Fengshan Xiong, Shi Yin.

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
