## [Decision Letter · Decision Letter 0]

2 Oct 2023

PONE-D-23-29519Green Credit Policy and Corporate Excess Cash HoldingsPLOS ONE

Dear Dr. Yin,

Thank you for submitting your manuscript to PLOS ONE. After careful consideration, we feel that it has merit but does not fully meet PLOS ONE’s publication criteria as it currently stands. Therefore, we invite you to submit a revised version of the manuscript that addresses the points raised during the review process.

We look forward to receiving your revised manuscript.

Kind regards,

Ercan Özen, PhD

Academic Editor

PLOS ONE

“This research received funding from the Hebei Provincial Department of Education University Humanities and Social Sciences Research Project "Research on Evaluation of Green Governance Effect of Environmental Pollution Liability Insurance for Listed Companies under the 'Double Carbon' Goal" (BJS2022003).

Additional Editor Comments:

Please arrange the paper according to reviewer comments.

Reviewers' comments:

Reviewer's Responses to Questions

**Comments to the Author**

1. Is the manuscript technically sound, and do the data support the conclusions?

Reviewer #1: Yes

Reviewer #2: Yes

2. Has the statistical analysis been performed appropriately and rigorously? 

Reviewer #1: Yes

Reviewer #2: Yes

3. Have the authors made all data underlying the findings in their manuscript fully available?

Reviewer #1: Yes

Reviewer #2: No

4. Is the manuscript presented in an intelligible fashion and written in standard English?

Reviewer #1: Yes

Reviewer #2: Yes

5. Review Comments to the Author

Reviewer #1: 1. Add discussion after results and conclusion.

2. implications and future scope of research are missing

3. discussions and limitations are missing

4. Sources below tables are not given

5. data availability statement is missing

Reviewer #2: 0 Abstract

Mention study period, novelty, and policy implications

1. Introduction

Latest literature review is required to identify research gap. Only two studies of 2021 are quoted. Include latest studies of 2022 and 2023.

Write the problem statement clearly.

2. Theoretical analysis and hypothesis formulation

To support the argument, include latest studies of 2022 and 2023. Why other factors are not taken into account that can reduce the excess cash holdings of enterprises. Proper justification is required

3. Research Design

this paper conducts 1% quantile tail reduction process for all continuous variables. Give reference.

Mention the total number of sample firms

Virtual variable? Use the word dummy variable.

In control variables, for age variable, usually Natural logarithm of the number of years plus 1 is taken? You are taking it without adding one. Give justification/ reference.

Operating activities per share… numerator should be divided into Number of shares, while in this paper it is divided into equity. Give justification/ reference.

For +β1×Posti,t×Treati,t, further explanation is required.

4. Empirical Results and Analysis

The median value of log (age) is 2.77, if we take the anti-log, it shows 589 years. Is suggest that half of the observation age is greater than 589 years, while half of the observation age is less than 589 years. How it is possible?

Author has made a good job by applying different models; however, discussion of results is weak. I is recommended that these analysis should be discussed comparatively and link it with prior studies and theories.

5. Research Conclusions and Implications

(1) Green credit policy are significantly negatively correlatedwiththe363 excess cash holdings of heavily polluting enterprises…. Rewrite the sentence, as this study is evaluating the impact, remove correlation word.

Include limitation and future research recommendations.

6. PLOS authors have the option to publish the peer review history of their article (what does this mean?). If published, this will include your full peer review and any attached files.

Reviewer #1: No

Reviewer #2: **Yes: **Prof. Dr. Muhammad Khalid Sohail

---

## [Author Response · Author response to Decision Letter 0]

16 Oct 2023

Dear Editors and Reviewers,

Thank you very much for your comments and professional advice. These opinions help to improve academic rigor of our article. Based on your suggestion and request, we have made corrected modifications on the revised manuscript. We hope that our work can be improved again. Furthermore, we would like to show the details as follows:

Reviewer #1: 

1. Add discussion after results and conclusion.

Response:

we have appropriately added the discussion of producing such results in the context of the research. Specifically, first, the discussion in section 4.2 Basic regression results and analysis is modified, as detailed on page6,line 222-226; Second, the discussion in section 4.4.1Test of action mechanism is modified, as detailed on page14,line 343-347; Third, the discussion in section 4.4.2Cross-section inspection is modified, as detailed on page15,line 364-372; and page16 line 385-394; Last, the discussion on the results of the 4.5Economic consequence test is revised, as detailed on page17,line 407-417.

2. implications and future scope of research are missing

Response:

At the end of the article, we have increased the limitations of the article and the possibility of future research, see pages 19, line 456-464.

3. discussions and limitations are missing

Response:

At the end of the article, we have increased the limitations of the article and the possibility of future research, see pages 19, line 456-464.

4. Sources below tables are not given

Response: We have added sources of information under the table，Mainly calculated by the author using stata.

5. data availability statement is missing

Response: We have increased data availability statement

Reviewer #2: 

0 Abstract

Mention study period, novelty, and policy implications

Response: We have increased the sample study period,novelty, and policy implications. The specific modification is on page 1, line 6-10,and line 16-21.

1. Introduction

Latest literature review is required to identify research gap. Only two studies of 2021 are quoted. Include latest studies of 2022 and 2023.

Write the problem statement clearly.

Response:

In the introduction, we reviewed the latest literature related to 2022 and 2023, as detailed on page 2, line 67-68 and line 79-80.

2. Theoretical analysis and hypothesis formulation

To support the argument, include latest studies of 2022 and 2023. Why other factors are not taken into account that can reduce the excess cash holdings of enterprises. Proper justification is required

Response:

Although there are many factors affecting cash holdings, we only focus on agency costs in this paper. According to previous literature analysis, green credit policies have stronger constraints on the financing of heavily polluting enterprises. According to the free cash flow hypothesis, the less available cash held by the management, the lower the agency costs, thus inhibiting the excess cash holdings of the management. The discussion of other factors can be continued in the future.

3. Research Design

(1)this paper conducts 1% quantile tail reduction process for all continuous variables. Give reference.

Response: We have added references. The specific modification is on page 3, line 149-151.

(2)Mention the total number of sample firms

Response: The total number of sample companies is 2340 . In the article we have added related descriptions. The specific modification is on page 3, line 148-149.

(3)Virtual variable? Use the word dummy variable.

Response: We have replaced Virtual with dummy, We made a change on page 4 ,line 166 &173 and page 8, line 258.

(4)In control variables, for age variable, usually Natural logarithm of the number of years plus 1 is taken? You are taking it without adding one. Give justification/ reference.

Response: We rechecked the data processing process and found that we did obtain the logarithm after+1 when conducting empirical research. However, due to our carelessness, we wrote the logarithm directly during the text writing process. Thank you very much to the reviewer. We made a change on page 4 ,line 192, Table 1.

(5)Operating activities per share… numerator should be divided into Number of shares, while in this paper it is divided into equity. Give justification/ reference.

Response: We rechecked the processing process of the original data, and indeed used the number of shares, rather than the equity, although the value of the equity and the number of shares are mostly the same in China's Shanghai and Shenzhen A-shares, and the number of shares was written as capital stock due to a clerical error. so we revised the definition of this variable and changed its denominator to the number of shares. Thank you very much for your careful review. The specific modification is on page 4, line 192,Table 1.

(6)For +β1×Posti,t×Treati,t, further explanation is required. 

Response: We explain post and treat in detail on page 4, line 173-184.

4. Empirical Results and Analysis

(1)The median value of log (age) is 2.77, if we take the anti-log, it shows 589 years. Is suggest that half of the observation age is greater than 589 years, while half of the observation age is less than 589 years. How it is possible?

Response:

The natural logarithm of age is used instead of the logarithm base 10, and the base number is the natural number e, e to the power of 2.7, which is about 14.87, which is basically consistent with the existing literature.

(2)Author has made a good job by applying different models; however, discussion of results is weak. I is recommended that these analysis should be discussed comparatively and link it with prior studies and theories.

Response: In the latest revised version, we have carried out appropriate discussion and analysis in accordance with the suggestions of reviewers, combining theories in each part. Specifically, first, the discussion in section 4.2 Basic regression results and analysis is modified, as detailed on page6,line 222-226; Second, the discussion in section 4.4.1Test of action mechanism is modified, as detailed on page14,line 343-347; Third, the discussion in section 4.4.2Cross-section inspection is modified, as detailed on page15,line 364-372; and page16 line 385-394; Last, the discussion on the results of the 4.5Economic consequence test is revised, as detailed on page17,line 407-417.

5. Research Conclusions and Implications

(1) Green credit policy are significantly negatively correlated with the

excess cash holdings of heavily polluting enterprises…. Rewrite the sentence, as this study is evaluating the impact, remove correlation word.

Include limitation and future research recommendations.

Response:

We have rephrased the sentence, rephrasing it as “Green credit policy significantly reduces the level of the excess cash holdings of enterprises. ”

At the end of the article, we have increased the limitations of the article and the possibility of future research, see pages 19, line 456-464.

Thank you very much for your attention and time. Look forward to hearing from you.

Best regards.

Jinhui Ning

E-mail: ningjinhui1120@163.com

---

## [Editor Report · Decision Letter 1]

25 Oct 2023

Green Credit Policy and Corporate Excess Cash Holdings

PONE-D-23-29519R1

Dear Dr. Shi Yin

We’re pleased to inform you that your manuscript has been judged scientifically suitable for publication and will be formally accepted for publication once it meets all outstanding technical requirements.

Kind regards,

Ercan Özen, PhD

Academic Editor

PLOS ONE
---

## [Editor Report · Acceptance letter]

31 Oct 2023

PONE-D-23-29519R1 

Green Credit Policy and Corporate Excess Cash Holdings 

Dear Dr. Yin:

I'm pleased to inform you that your manuscript has been deemed suitable for publication in PLOS ONE. Congratulations! Your manuscript is now with our production department. 

Kind regards, 

on behalf of

Dr. Ercan Özen 

Academic Editor

PLOS ONE